# Response of stratospheric water vapor and ozone to the unusual timing of El Niño and QBO disruption in 2015–2016

Mohamadou Diallo[1,2], Martin Riese[1], Thomas Birner[3], Paul Konopka[1], Rolf Müller[1], Michaela I. Hegglin[4], Michelle L. Santee[5], Mark Baldwin[6], Bernard Legras[2], and Felix Ploeger[1]

[1]Institute of Energy and Climate Research, Stratosphere (IEK-7), Forschungszentrum Jülich, 52 425 Jülich, Germany.
[2]Laboratoire de Météorologie Dynamique, UMR8539, IPSL, UPMC/ENS/CNRS/Ecole Polytechnique, Paris, France.
[3]Department of Atmospheric Science, Colorado State University, Boulder, CO, USA.
[4]Department of Meteorology, University of Reading, Reading, UK.
[5]Jet Propulsion Laboratory, California Institute of Technology, Pasadena, California, USA.
[6]College of Engineering, Mathematics and Physical Sciences, University of Exeter, Exeter, UK.

**Correspondence:** Mohamadou Diallo (m.diallo@fz-juelich.de)

**Abstract.**

The stratospheric circulation determines the transport and lifetime of key trace gases in a changing climate, including water vapor and ozone, which radiatively impact surface climate. The unusually warm El Niño Southern Oscillation (ENSO) event aligned with a disrupted Quasi–Biennial Oscillation (QBO) caused an unprecedented perturbation to this circulation in 2015–2016. Here, we quantify the impact of the alignment of these two phenomena in 2015–2016 on lower stratospheric water vapor and ozone from satellite observations. We show that the warm ENSO event substantially increased water vapor and decreased ozone in the tropical lower stratosphere. The QBO disruption significantly decreased global lower stratospheric water vapor and tropical ozone from early spring to late autumn. Thus, this QBO disruption reverses the lower stratosphere moistening triggered by the alignment of the warm ENSO event with westerly QBO in early boreal winter. Our results suggest that the interplay of ENSO events and QBO phases will be crucial for the distributions of radiatively active trace gases in a changing future climate, when increasing El Niño-like conditions and decreasing lower stratospheric QBO amplitude are expected.

## 1  Introduction

The lower stratosphere (10–25 km) is a key region in a changing climate. Transport, mixing, and chemistry in this region regulate the amount of key greenhouse gases, such as water vapor and ozone, which radiatively impact temperatures both locally (e.g. Forster and Shine, 2002) and globally (e.g. Forster and Shine, 1999; Solomon et al., 2010; Riese et al., 2012; Dessler et al., 2013). Ozone is mainly produced in the stratosphere (10–50 km) and is directly regulated in the tropical lower stratosphere by the upwelling strength of the stratospheric circulation (Randel et al., 2007; Abalos et al., 2013). Conversely, water vapor mainly originates from the troposphere and its stratospheric concentration is controlled by the tropical cold point tropopause temperatures (Holton and Gettelman, 2001; Hu et al., 2016) and production from methane oxidation (le Texier et al., 1988; Dessler et al., 1994). The amount of stratospheric water vapor is thereby modulated by the coldest temperatures experienced by air parcels ascending through the tropical tropopause layer (TTL) (e.g. between 14–19 km, Fueglistaler et al.,

2009; Fueglistaler, 2012; Schoeberl and Dessler, 2011). The dehydration in the air parcels crossing through the TTL plays an important role in the control of the lower stratospheric moisture. Stratospheric water vapor is the primary source of stratospheric hydrogen oxide radicals, which are driving important gas-phase ozone loss cycles, and also strongly influences heterogeneous chemistry on cold sulphate aerosol and the formation of polar stratospheric clouds, which promote chlorine activation and

polar ozone loss (e.g. Solomon et al., 1986; Manney et al., 1994; Crutzen et al., 1995; Müller et al., 1997; Solomon, 1999; Kirk-Davidoff et al., 1999; Dvortsov and Solomon, 2001; Drdla and Müller, 2012).

Water vapor and ozone abundances in the tropical lower stratosphere show multi-timescale variations ranging from daily to decadal (e.g. Randel et al., 2004; Fueglistaler and Haynes, 2005; Fujiwara et al., 2010; Hegglin et al., 2014) dominated by temperature variations and the tropical upwelling strength, respectively (e.g. Randel et al., 2007; Rosenlof and Reid, 2008;

Randel et al., 2010; Fueglistaler et al., 2013; Randel and Jensen, 2013). These temperature fluctuations are driven by the varying strength of the stratospheric circulation. Beyond the annual cycle (tape recorder) (Mote et al., 1996; Glanville and Birner, 2017), one key driver of the interannual variability of water vapor is the interaction between the El Niño-Southern Oscillation (ENSO) and the Quasi–Biennial Oscillation (QBO) (Garfinkel and Hartmann, 2007; Taguchi, 2010), which, in turn, modulates the stratospheric circulation.

The stratospheric mean meridional circulation is the Brewer-Dobson circulation (BD-circulation) (e.g. Brewer, 1949; Butchart, 2014), defined as a slow circulation in which air parcels rising in the tropics drift poleward into the stratosphere and are transported downward in the high-latitude regions via its shallow and deep branches (Birner and Bönisch, 2011; Bönisch et al., 2011). Driven by wave breaking in the stratosphere (Haynes et al., 1991; Rosenlof and Holton, 1993; Newman and Nash, 2000; Plumb, 2002), the BD-circulation varies on subseasonal to decadal timescales.

The QBO is a major mode of variability of the tropical upwelling of the BD-circulation (Lindzen and Holton, 1968; Plumb and Bell, 1982). The QBO is composed of alternating westerly and easterly zonal wind shears, descending in the tropical stratosphere with a period of ∼28 months. Mostly driven by equatorially trapped waves (Wallace et al., 1993; Baldwin et al., 2001; Ern and Preusse, 2009; Ern et al., 2014), the QBO triggers a modulation of vertical and meridional transport in the stratosphere by affecting temperature and heating rates (Niwano et al., 2003; Punge et al., 2009). The easterly shear is associated

with enhanced tropical upwelling and anomalously cold tropopause temperatures. As the easterly shear reaches the tropopause, it therefore causes low anomalies of tropical lower stratospheric water vapor and ozone. Conversely, the westerly shear reduces the tropical upward motion, but also enhances the horizontal transport and mixing of stratospheric trace gases and aerosols poleward (Plumb and Bell, 1982; Trepte and Hitchman, 1992). The tropical upwelling is anti-correlated with the tropical temperature above the tropopause and its strength determines stratospheric ozone and water vapor entry values by modulating

TTL temperatures (Yulaeva et al., 1994; Randel et al., 2006; Flury et al., 2013).

Another major mode of climate variability that affects the variability of the BD-circulation is the ENSO. ENSO is a coupled atmosphere-ocean phenomenon covering the equatorial Pacific Ocean with drastic changes in regional sea surface temperatures (SSTs), impacting surface weather and climate (e.g. Bjerknes, 1969; Cagnazzo and Manzini, 2009; Wang et al., 2016). ENSO alternates between anomalously warm (El Niño) and cold (La Niña) conditions in the tropical Pacific Ocean at intervals of

2-8 years (Philander, 1990; Baldwin and O'Sullivan, 1995). In addition to warming the troposphere, El Niño events cool

the tropical lower stratosphere and strengthen the tropical upwelling of the BD-circulation, decreasing ozone in the tropical lower stratosphere (Randel et al., 2009). From a zonal mean perspective, El Niño events induce tropospheric warming and stratospheric cooling with a node near the tropopause (Randel et al., 2009; Mitchell et al., 2015). Stratospheric water vapor, however, is predominantly controlled by cold point temperatures over the tropical Wester Pacific (Hu et al., 2016). El Niño events are associated with warmer cold point temperatures over this region, thereby, causing increased lower stratospheric water vapor. In contrast, La Niña events induce an opposite effect (e.g. Calvo et al., 2010; Konopka et al., 2016).

Climate models predict that increasing greenhouse gas levels will speed up the mean tropical upwelling of the BD-circulation in the future (McLandress and Shepherd, 2009; Garny et al., 2011; Lin and Fu, 2013; Butchart, 2014; Hardiman et al., 2014). A previous study finds a long-term decrease of the QBO amplitude in the lowermost stratosphere associated with this strengthening tropical upwelling (Saravanan, 1990), consistent with projections of global climate models (Kawatani et al., 2011; Kawatani and Hamilton, 2013). Future projections of climate models also predict a shift of the basic state toward more frequent El Niño conditions in a warming climate (Timmermann et al., 1999; van Oldenborgh et al., 2005; Latif and Keenlyside, 2009; Cai et al., 2014). In this context, it is of particular importance to better understand the impact of the interplay between ENSO and QBO on changes in stratospheric water vapor and ozone (Solomon et al., 2010; Riese et al., 2012), which directly impact the global radiative forcing of climate (Forster and Shine, 1999; Butchart and Scaife, 2001).

## 2   Puzzling water vapor anomalies in 2015–2016

Recently, a previously unobserved timing of this interplay between ENSO and QBO occurred. During the boreal winter 2015–2016, a strong El Niño event (among the 3 strongest El Niño events on the record) (Huang et al., 2016) was aligned with a westerly QBO phase. This westerly QBO phase was abruptly disrupted well before completion by an easterly phase in January 2016 (Osprey et al., 2016; Newman et al., 2016). The interplay of both circulation anomalies caused large changes in trace gas transport, the climate implications of which are currently a topic of debate. Based on modern reanalyses and satellite observations, including Aura Microwave Limb Sounder (MLS) water vapor mixing ratios, Avery et al. (2017) argued that the most recent El Niño event significantly moistened the lower stratosphere ($\pm 0.9$ ppmv) due to particularly warm tropopause temperature anomalies in the tropical Western Pacific. Using a simple linear regression of MLS water vapor mixing ratios at 82 hPa with a QBO index at 70 hPa, Avery et al. (2017) concluded that the contribution of the QBO disruption was small (up to 0.1 ppmv) at 82 hPa, even though the study mainly focused on ENSO, in particular, the role of tropical convective cloud ice in stratospheric hydration. In contrast, Tweedy et al. (2017) mainly focused on QBO disruption impact and attributed changes in the global stratospheric water vapor content from spring to autumn to the QBO disruption during the 2015–2016 winter. However, Tweedy et al. (2017) also acknowledged that the strong El Niño event could have strongly influenced their correlation (composite) analyses based on MLS satellite and radiosonde observations. Disentangling the effects of ENSO and QBO on this anomalous trace gas variability and identifying the dominant driver of recent lower stratospheric water vapor changes during 2015–2016 is a challenging task. A detailed explanation of the reasons for this lower stratospheric water vapor variability in 2015–2016 is still lacking.

Here, we quantify the impact of the interaction between the most recent El Niño event and the QBO disruption on entire lower stratospheric ozone and water vapor from spaceborne measurements during the 2015–2016 period. We describe the satellite observational data record and multiple regressions in Section 3. Section 4 describes the anomalous stratospheric circulation in boreal winter of 2015–2016 and Section 5 shows evidence for the impact of the El Niño event and QBO disruption on stratospheric ozone and water vapor. Finally, we discuss our results in the context of the puzzling water vapor response to the interaction of these two phenomena.

## 3 Data and Methodology

The data analysed here are monthly mean ozone ($O_3$) and water vapor ($H_2O$) mixing ratios in the lower stratosphere from the Aura Microwave Limb Sounder satellite observations covering the period 2005–2016 (Livesey et al., 2017). The MLS instrument, flying aboard the EOS-Aura satellite, is designed to measure a wide range of physical and chemical quantities, including $O_3$ and $H_2O$ (Waters et al., 2006). The version 4.2 MLS data were produced with improved retrieval algorithms, which substantially reduced the occurrence of unrealistically small $O_3$ values at 215 hPa in the tropics observed in the previous version 2.2 MLS product (Livesey et al., 2008). Note that the version 4.2 MLS data used here are not significantly different from the previous version MLS observations at pressures less than 100 hPa, but show less oscillatory behavior and fewer retrieval artifacts induced by cloud contamination in the tropical upper troposphere-lower stratosphere (UTLS). The vertical resolution, precision, systematic uncertainty and lowest recommended vertical range of the relevant v4.2 data are respectively 2.5–3 km, $\pm10$–40%, $\pm10$–25% and 316 hPa for $H_2O$ and 3–3.5 km, $\pm0.02$–0.04 ppmv, $\pm0.02$–0.05 ppmv+ $\pm5$-10% and 261 hPa for $O_3$ for individual profile measurements with a spatial representativeness of $\sim$200–300 km along the orbital-track line of sight (Schwartz et al., 2013; Livesey et al., 2017; Santee et al., 2017). The regression results will be not affected by these intrinsic uncertainties since they apply to the $H_2O$ and $O_3$ mixing ratios and not the anomalies. In addition, Hegglin et al. (2013) show that MLS zonal monthly mean $H_2O$ show very good to excellent agreement with the Multi-Instrument Mean (MIM) in comparison between 13 instruments, throughout most of the atmosphere (including the UTLS) with mean deviations from the MIM between +2.5 and +5%, making these random errors irrelevant for the averaged monthly zonal mean $H_2O$ anomalies used in this study. Additional detailed information on the quality of $O_3$ and $H_2O$ in the upper troposphere–stratosphere in previous versions can be found in dedicated validation papers (Read et al., 2007; Lambert et al., 2007; Livesey et al., 2008; Froidevaux et al., 2008).

As an illustration of the robustness of the regression results, MLS water vapor is compared in Sect. 6 to simulated $H_2O$ from the Chemical Lagrangian Model of the Stratosphere (CLaMS) (McKenna et al., 2002; Konopka et al., 2004). Lagrangian transport in CLaMS is based on 3D backward trajectories and a parameterization of small-scale mixing, which relates mixing to deformations in the large-scale flow. The model uses an isentropic vertical coordinate, with vertical transport driven by the total diabatic heating rate (Ploeger et al., 2010). The model simulations considered for this paper are driven by temperatures, horizontal winds and diabatic heating rates from European Centre for Medium-range Weather Forecasts (ECMWF) ERA-Interim reanalysis (Dee et al., 2011). For the wind and temperature fields, CLaMS uses the native ERA-Interim vertical resolution,

therefore, has higher vertical resolution than MLS. The mean vertical resolution of air parcels in CLaMS Lagrangian model is about 400 m near the tropopause. Stratospheric water vapour in CLaMS is calculated based on a simplified dehydration scheme, which is based on freezing at 100% saturation and a parameterized ice particle fall-out (e.g., Poshyvailo et al., 2018), and additional chemical production in the middle stratosphere due to methane oxidation. For further details about the model

set-up used here see Pommrich et al. (2014). Based on comparison of modern reanalysis intercomparisons, Long et al. (2017) show that the ERA-Interim temperatures compare favorably to other reanalyses throughout most the atmosphere, including the TTL region. The assimilation of Global Positioning System radio occultation data since December 2006 have reduced the ERA-Interim cold temperature bias compared with radiosondes in the tropopause layer and the lower stratosphere (Poli et al., 2010). ERA-Interim tropical tropopause temperatures have also been shown to compare very well against in situ observations

over the eastern tropical Pacific (Ueyama et al., 2014). Based on comparison of ERA-Interim tropopause temperature with in situ balloon observations, Podglajen et al. (2014) found fairly good agreement with a weak positive bias of 0.6 K and a standard deviation of 1.8 K in the TTL. Schoeberl et al. (2012) show that even small temperature differences between reanalyses and observations can still induce differences in the associated $H_2O$ saturation mixing ratio using a trajectory model driven by ERA-interim. However, the CLaMS dehydration scheme has been shown to provide lower stratospheric $H_2O$ anomalies in

good agreement with current satellite observations, including the MLS product, giving good confidence in the CLaMS $H_2O$ reconstruction from the large-scale perspective (e.g., Ploeger et al., 2013; Tao et al., 2015; Lossow et al., 2018). In addition, the $H_2O$ anomaly time series is less affected by biases, which do not have ENSO or QBO signals.

To disentangle the ENSO and QBO impact on these stratospheric trace gases from the other sources of natural variability, the 2005–2016 monthly zonal mean $O_3$ and $H_2O$ mixing ratios from MLS observations are analysed as a function of latitude

($\phi$) and altitude (z) using a multiple regression model. This regression method is an established method and appropriate to disentangle the relative influences of the considered climate indices on stratospheric trace gas variability, as it includes time lag coefficients for both QBO and ENSO. For more details about the method and its further applications see Diallo et al. (2012, 2017). The regression method decomposes the temporal evolution of the monthly zonal mean trace gas mixing ratio, $\chi$, in terms of a long-term linear trend, seasonal cycle, QBO, ENSO, Aerosol Optical Depth (Vernier et al., 2011) and a residual.

The model yields for a given trace gas, $\chi$ (herein $O_3$ and $H_2O$)

$$\chi(t,\phi,z) = a(\phi,z) \cdot t + C(t,\phi,z) + \sum_{k=1}^{3} b_k(\phi,z) \cdot P_k(t - \tau_k(\phi,z)) + \varepsilon(t,\phi,z) \tag{1}$$

where $P_k$ represents the predictors or proxies. $P_1$ is a normalised QBO index (QBOi) from CDAS/Reanalysis zonally averaged winds at 50 hPa, $P_2$ is the normalised Multivariate ENSO Index (MEI) (Wolter and Timlin, 2011) and $P_3$ is the AOD from satellite data (Vernier et al., 2011). The coefficients are a linear trend $a$, the annual cycle $C(t,\phi,z)$, the amplitude $b_1$ and the lag

$\tau_1(\phi,z)$ associated with the QBO, the amplitude $b_2$ and the lag $\tau_2(\phi,z)$ associated with ENSO and the amplitude $b_3$ and the lag $\tau_3(\phi,z)$ associated with AOD. The constraint applied to determine the parameters $a$, $b_1$, $b_2$, $b_3$, $\tau_1(\phi,z)$, $\tau_2(\phi,z)$, $\tau_3(\phi,z)$ and $C$ is to minimise the residual $\varepsilon(t,\phi,z)$ in the least squares sense. Because of the presence of lags in the QBO, ENSO and AOD terms in equation (1), the problem is nonlinear and the residual may have multiple minima as a function of the parameters.

In order to determine the optimal values of $\tau_1(\phi, z)$, $\tau_2(\phi, z)$ and $\tau_3(\phi, z)$, the residual is first minimised at fixed lag and then selected from a range of possible lags. This is done in sequence for QBO, ENSO and AOD. Here we neglect solar forcing, because our data set covers only one solar period. Uncertainty estimates for the statistical fits are calculated using a Student's t-test technique (Zwiers and von Storch, 1995; Bence, 1995; von Storch and Zwiers, 1999).

## 4  Anomalous stratospheric circulation in the 2015–2016 boreal winter

Almost simultaneously with the exceptionally strong El Niño in boreal winter of 2015–2016 (Huang et al., 2016), the fairly regular QBO cycle was disrupted by an unexpected shift from westerly (positive QBOi) to easterly (negative QBOi) winds. In January 2016, an easterly phase developed in the center of the westerly phase, breaking the regular cycle of easterly–westerly phase (Osprey et al., 2016; Newman et al., 2016). The QBO disruption was attributed to planetary Rossby waves propagating from the northern hemisphere to the southern hemisphere in the winter stratosphere (Osprey et al., 2016; Coy et al., 2017; Hitchcock et al., 2018), potentially triggered by the strong El Niño event (Schirber, 2015; Dunkerton, 2016; Christiansen et al., 2016; Barton and McCormack, 2017). Together, the most recent El Niño event and the QBO disruption are expected to impact the tropical upwelling, via wave–mean flow interaction (Holton, 1979; Dunkerton, 1980; Grimshaw, 1984) and control of the cold point temperatures (Kim and Son, 2012; Kim and Alexander, 2015), therefore affecting the transport and distribution of stratospheric trace gases (Avery et al., 2017; Tweedy et al., 2017).

Figures 1(a, b) show the interannual variability of the deseasonalised $O_3$ (a) and $H_2O$ (b) in the tropical lower stratosphere as a percentage change relative to the monthly mean mixing ratio during the 2005–2016 period. Particularly, during the 2015–2016 period, the deseasonalised $O_3$ shows negative anomalies in the lower stratosphere (380–550 K) as expected due to the enhanced tropical upwelling caused by both the extreme El Niño event and the QBO disruption (e.g. easterly wind shear at 100–40 hPa). In contrast, the $H_2O$ variability (tape recorder) is more challenging to interpret because of its regulation by the tropical cold point tropopause temperatures. The complexity in $H_2O$ variability lies in its dependency on ENSO, on the QBO phases (Liess and Geller, 2012), seasons (early or late in the winter) and location (Central or Eastern Pacific, where the ENSO maximum occurs (Garfinkel et al., 2013)). Therefore, to elucidate the ENSO and QBO impact on the stratospheric $O_3$ and $H_2O$ anomalies, the multiple regression is performed both without and with explicitly including ENSO and QBO signals to isolate the impact of the ENSO and QBO on these trace gases, respectively. The difference between the residual ($\varepsilon$ in Eq. (1)) with and without explicit inclusion of the ENSO and QBO signals gives the ENSO- and QBO-induced impact on stratospheric $O_3$ and $H_2O$ anomalies. This approach of differencing the residuals is similar to direct calculations, projecting the regression fits onto the ENSO and QBO basis functions i.e. the ENSO and QBO predictor time series (see supplement Fig. 2 and 4 in Diallo et al. (2017)). In addition, this differencing approach avoids the need to reconstruct the time series after the regression analysis.

## 5 Results

### 5.1 Impact of the 2015–2016 El Niño on lower stratospheric $O_3$ and $H_2O$

Figures 2(a, b) show time series of the ENSO-induced variability in tropical monthly mean $O_3$ and $H_2O$ estimated from the difference between the residual ($\varepsilon$ in Eq. (1)) without and with explicit inclusion of the ENSO signal for the 2005–2016 period. Figure 2a indicates that the most recent El Niño event produces an extremely large negative $O_3$ anomaly in the lower stratosphere, inducing a record minimum anomaly of minus 15% in the tropics, consistent with previous studies (Randel et al., 2009; Calvo et al., 2010; Konopka et al., 2016). This strong decrease in $O_3$ mixing ratio is interpreted as a strengthening of the tropical upwelling induced by El Niño (Randel et al., 2009). In addition, by effectively warming the cold point temperature (Hu et al., 2016), the recent strong El Niño event in 2015–2016 regulates the stratospheric $H_2O$ entry mixing ratio by significantly inducing positive anomalies in the tropical lower stratosphere between 380–450 K (Fig. 2b). These changes in $H_2O$ mixing ratio in the TTL reach 10–15% and are consistent with a recent study (Avery et al., 2017).

Figures 2(c, d) depict the zonal mean impact of the recent strong El Niño on $O_3$ (c) and $H_2O$ (d) calculated from the difference between the residuals, which is similar to Fig. 2(a, b) but averaged for the 2015–2016 period. Figure 2c shows that the $O_3$ mixing ratio decreases throughout the tropics during El Niño as expected due to the enhanced tropical upwelling, bringing air poor in $O_3$ from the troposphere. In the extratropics (poleward of 30°N) of the northern hemisphere, there is a related increase of $O_3$ mixing ratios due to enhanced downwelling from the shallow branch of the BD-circulation (Neu et al., 2014). The negative $O_3$ anomalies seen in the southern hemisphere polar region are likely a consequence of the Antarctic ozone hole during the austral spring (Solomon, 1999; WMO, 2014).

Clearly, there is a strong increase in $H_2O$ anomalies in the lower stratosphere related to the extreme El Niño event from February 2015 to December 2016 (Figure 2d), which induced generally warmer tropical cold point tropopause temperatures (Hu et al., 2016). These positive $H_2O$ anomalies are consistent with the known effect of El Niño to moisten the tropical lower stratosphere (e.g., Bonazzola and Haynes, 2004; Randel et al., 2004; Fueglistaler et al., 2005; Konopka et al., 2016). The induced $H_2O$ anomalies by the strong El Niño event propagate toward the extratropical lower stratosphere. This propagation is likely attributable to the horizontal transport caused by the shallow branch of the residual circulation near the subtropics and by eddy mixing at higher latitudes, poleward of about 50°N (Hegglin and Shepherd, 2007; James and Legras, 2009; Ploeger et al., 2013). The largest $H_2O$ anomalies occur between 20–50° S/N near the subtropical jet due to the convection shift (L'Heureux et al., 2017; Avery et al., 2017) and in the upper troposphere. The positive H2O anomalies associated with El Niño below $\sim$400 K are related to the extended tropospheric moist anomaly (Fig. 2d), which is partly associated with an upward shifting tropopause (Randel et al., 2004; Lorenz and DeWeaver, 2007; Lu et al., 2008) and partly due to smearing effect arising from the limited 2.5-3 km vertical resolution of the MLS $H_2O$ measurements. Using high resolution temperature data and climate model simulations, Randel et al. (2009) showed that there is a clear separation between a warming troposphere and cooling lower stratosphere for the zonal average ENSO signal, with a node near the tropical cold point tropopause (i.e. a demarcation between the warming and cooling regime). However, zonal mean $H_2O$ anomalies do not exactly follow the zonal mean temperature, but critically depend on the geographical distribution of lowest temperature regions (Bonazzola and Haynes, 2004; Konopka

et al., 2016). Konopka et al. (2016) argued that El Niño causes colder zonal mean temperatures, but also warmer temperatures over the West Pacific region, which is most critical for stratospheric entry water vapor (e.g., Fueglistaler et al., 2004). As a net effect, zonal mean $H_2O$ mixing ratios turn out to be larger during El Niño than La Niña.

With the exception of the Antarctic polar vortex, the $H_2O$ anomalies above 450 K become negative over the entire strato-
sphere, with a minimum occurring in the inner tropics between 450–550 K. These negative $H_2O$ anomalies are related to air which entered the stratosphere before the onset of El Niño and a related upward propagating tape-recorder signal.

## 5.2  Impact of the QBO disruption on lower stratospheric $O_3$ and $H_2O$

Figures 3(a, b) show time series of the QBO-induced variability in tropical monthly mean $O_3$ and $H_2O$ estimated from the difference between the residual ($\varepsilon$ in (1)) without and with explicit inclusion of the QBO signal for the 2005–2016 period. For
the QBO-induced impact, anomalies in both trace gases are roughly in phase below 500 K, with a delay of a few months for the $H_2O$ anomalies. Both trace gases reveal a footprint of the QBO disruption in their anomalies, e.g. a shift from increasing mixing ratios (positive anomalies) related to the westerly wind shear (positive QBOi) to decreasing mixing ratios (negative anomalies) related to the easterly wind shear (negative QBOi). The occurrence of the easterly wind shear at 40 hPa ($\sim$550 K) induces significant negative $O_3$ and $H_2O$ anomalies as large as 15–20% between 380–450 K consistent with upward transport
of young and dehydrated air poor in $O_3$ and $H_2O$ into the lower stratosphere (Fig. 3). The response of the $O_3$ anomalies to the QBO shift is sudden and follows the monthly mean zonal mean wind changes as represented in ERA-Interim reanalysis. The $H_2O$ response to the QBO disruption is delayed by about 3–6 months due to its tropospheric origin, and reaches its minimum value in autumn 2016. The results for both $O_3$ and $H_2O$ are consistent with those shown previously by Tweedy et al. (2017). The westerly wind shear that appears between 30–10 hPa ($\sim$570–600 K) reduces the upward motion of the BD-circulation and
causes positive $O_3$ and $H_2O$ anomalies of up to 5% and 10% in the lower stratosphere (above 570 K) during the early boreal winter of 2015–2016.

The zonal mean impact of the QBO disruption on $O_3$ and $H_2O$ anomalies is calculated as the difference between the residuals averaged between April–December 2016 (Figure 3(c, d), respectively). In the tropics, the observed negative $O_3$ anomalies in Fig. 3a reach up to 450 K due to the easterly QBO phase, whilst above that level, the positive $O_3$ anomalies remain mainly
confined below 600 K due to the westerly QBO phase (Fig. 3c). In the extratropics, the changes in $O_3$ anomalies reflect large variability at high latitudes, which can be associated with the effect of the QBO influence on the extratropical circulation (Hampson and Haynes, 2006; Damadeo et al., 2014), stratospheric major warmings and chemical processes (WMO, 2014; Manney and Lawrence, 2016).

In contrast to the strong El Niño, the QBO disruption significantly dehydrates the lower stratosphere (Fig. 3d). Below the
450 K level, the lower stratospheric $H_2O$ abundances globally decrease due to the enhanced tropical upwelling and related decrease of cold point temperature (Jensen et al., 1996; Hartmann et al., 2001; Geller et al., 2002; Schoeberl and Dessler, 2011). This decrease in $H_2O$ mixing ratios reaches a maximum net change of about minus 10–20% (Fig. 3d). The strongly dehydrated air rising through the tropical tropopause propagates more toward the northern hemisphere than southern hemisphere because of the asymmetry of the meridional circulation driven by planetary wave activity (Holton and Gettelman, 2001; Flury et al., 2013;

Konopka et al., 2015) and eddy mixing (Haynes and Shuckburgh, 2000; Nakamura, 2001; Hegglin et al., 2005). The large-amplitude negative $H_2O$ anomalies at high latitudes are likely due to the large atmospheric variability in that region, which is related to stratospheric major warmings and chemical processes (WMO, 2014; Manney and Lawrence, 2016), or the high-latitude influence of the QBO (Holton and Tan, 1980; Baldwin and Dunkerton, 1998; Anstey and Shepherd, 2014). The zonal mean picture of decreasing $H_2O$ related to the QBO disruption is consistent with the findings of Tweedy et al. (2017), which suggested a global dehydration of the lower stratosphere. The positive $H_2O$ anomalies with a maximum occurring between 500–550 K are related to the effect of the preceding westerly QBO phase on TTL temperatures and the upward propagating tape-recorder signal.

## 6   Discussion

Two previous studies (i.e. Avery et al. (2017) and Tweedy et al. (2017)) focussing on ENSO and QBO, respectively, led to puzzling $H_2O$ anomalies in 2015–2016. Avery et al. (2017) argued that the most recent El Niño event significantly moistened the lower stratosphere, with the QBO disruption having only a small contribution. In contrast, Tweedy et al. (2017) attributed the lower stratospheric $H_2O$ changes from spring to autumn to the QBO disruption. Our analysis shows that the QBO disruption significantly decreased global lower stratospheric $H_2O$ from early spring to late autumn and reversed the lower stratosphere moistening triggered by the alignment of the warm ENSO event with westerly QBO in early boreal winter. These presented regression results are significant with respect to the measurement uncertainties.

An interesting open question concerns what would have happened to the lower stratospheric $H_2O$ anomalies if there had been no QBO disruption? The clearest picture emerges from the latitude–time series of $H_2O$ anomalies in Fig. 4, on which we concentrate our discussion in the following. Figure 4 shows the deseasonalised time series (a) together with the impact of the QBO (b) and ENSO (c) on $H_2O$ averaged in the lower stratosphere between 380–425 K. Remarkably, the variability in $H_2O$ anomalies shown in Fig 4a is largely explained by the interplay between the ENSO and QBO induced variability. In early boreal winter 2015–2016, Figure 4a shows that the lower stratosphere was strongly moistened by both the strong El Niño event (Fig. 4b) and the westerly QBO phase (Fig. 4c). Considered as one of the three strongest occurring since 1950 (Huang et al., 2016; Hu et al., 2016), the most recent El Niño event stands out in the decadal record of ENSO impact on $H_2O$ in the lower stratosphere (see black vertical dashed-line in Fig. 4b), consistent with the findings of (Avery et al., 2017). The positive $H_2O$ anomalies induced by this most recent El Niño slowly propagate with time into the extratropical lower stratosphere of both hemispheres due to the shallow branch of BD-circulation and eddy mixing processes. During the boreal winter 2015–2016 (DJFM, December–March), the westerly QBO phase contribution to $H_2O$ anomalies adds to the El Niño-induced $H_2O$ variability, resulting in particularly large $H_2O$ anomalies, consistent with the findings of Tweedy et al. (2017).

However, the QBO shift from westerly to easterly wind shear at 40 hPa ($\sim$550 K) suddenly reverses the extreme lower stratospheric moistening by significantly decreasing $H_2O$ from boreal spring 2016 to boreal winter 2016–2017 (Fig. 4c). The QBO disruption contributes the most to the lower stratospheric water budget between 380–425 K, with strong negative $H_2O$ anomalies of about 20% from boreal spring to boreal winter 2016–2017 compared to the El Niño, which only induces about

5–10% increase on average in this layer during the same period. Therefore, if there had been no QBO disruption during the boreal winter of 2015–2016 with an ongoing westerly QBO phase, the $H_2O$ anomalies would have likely increased to more than 25%, leading to changes larger than previously observed in the lower stratospheric water budget.

The control of the interannual variability in lower stratospheric $H_2O$ anomalies critically depends on the alignment of the ENSO events and QBO phases. Alignment of a westerly QBO phase with El Niño leads to strongly positive $H_2O$ anomalies as illustrated during the boreal winters of 2006–2007 and 2015–2016. Alignment of an easterly QBO phase with La Niña induces strongly negative $H_2O$ anomalies, for example, as seen during the boreal winter of 2005–2006 and 2007–2008. This result is consistent with previous studies based on observations (Yuan et al., 2014) and climate models (Brinkop et al., 2016). According to the findings of Yuan et al. (2014), the greatest dehydration of air entering the stratosphere from the troposphere occurs during the winter under La Niña and easterly QBO phase. Brinkop et al. (2016) suggested that a large decline in $H_2O$ anomalies can be found after strong El Niño/La Niña events combined with a transition from the westerly QBO phase during La Niña to the easterly QBO phase. In conclusion, the alignment of the westerly QBO phases with El Niño events (e.g. 2006–2007, early 2015–2016) and easterly QBO with La Niña events (e.g. 2005–2006, 2007–2008) are the key factors in creating extreme lower stratospheric water vapor anomalies via a control of cold point tropopause temperatures. Consistent with this picture, the variance of the deseasonalised $H_2O$ time series is largely captured by this interplay of the ENSO events and QBO phases as shown in Fig. 5. The variance of the QBO and ENSO induced changes in $H_2O$ anomalies shows that the QBO contributes the largest part to the $H_2O$ variability (Fig. 5).

In addition, when the ENSO signal is weak or moderate (e.g. 2012–2015, early winter 2016–2017), the lower stratospheric $H_2O$ anomalies are dominated by the QBO phases. This QBO control of the lower stratospheric $H_2O$ budget is also illustrated during the boreal winter of 2010–2011. Despite the ongoing La Niña event, which dehydrated the lower stratosphere, the impact of the westerly QBO phase on $H_2O$ anomalies dominated, leading to positive anomalies approaching 25%. According to Nedoluha et al. (2015), the westerly wind shear persisted slightly longer than usual during the 2008–2013 period (QBOi in Fig. 4c). Therefore, this persistence of the westerly QBO can explain the large $H_2O$ anomalies during this period. Note that the 2011 winter had an extreme anomalously strong vortex, i.e. strongly reduced BD-circulation, which also might have contributed to these large positive anomalies (Manney et al., 2011). An additional example of this QBO control on the lower stratospheric $H_2O$ anomalies is the drop in $H_2O$ during the 2012–2013 boreal winter (Urban et al., 2014). These extremely negative $H_2O$ anomalies are associated with the rapid cooling of the tropical cold point tropopause temperatures induced by easterly wind shear and a major sudden stratospheric warming (Evan et al., 2015; Tao et al., 2015). This cooling of the tropical cold point tropopause temperatures is induced by a downward shift of the zero wind line (∼30 hPa) during easterly wind shear, inducing more subtropical wave dissipation at low latitudes, therefore efficiently speeding up the shallow branch of the BD-circulation (Garny et al., 2011; Gómez-Escolar et al., 2014). Therefore, based on these recent findings (Gómez-Escolar et al., 2014; Evan et al., 2015; Tao et al., 2015), we can explain the sudden drop in the lower stratospheric moistening from boreal spring 2016 to boreal winter 2016–2017 despite the strong El Niño as a consequence of the rapid cooling of the tropical cold point tropopause temperatures induced by the QBO disruption (easterly) (Tweedy et al., 2017) and the major stratospheric final warming in 2016 (Manney and Lawrence, 2016), which strengthened the shallow branch of the BD-circulation.

In order to gain confidence in the robustness of the above discussed results and to illustrate the ability of the CLaMS model to capture the unusual timing of QBO shift and El Niño in 2015–2016, we have also estimated the impact of their interplay on lower stratospheric $H_2O$ anomalies from the CLaMS simulations using the same regression method. Consistently, the CLaMS $H_2O$ anomalies show characteristics in good agreement with the zonally averaged $H_2O$ anomalies from MLS (Fig. 6a). CLaMS simulations and MLS observations agree remarkably well throughout the entire record and especially the El Niño and QBO signals in 2015–2016 (Fig. 6(b, c)). In particular, also in the model the El Niño signal is much weaker than the impact of the QBO disruption on lower stratospheric $H_2O$. The influence of the QBO disruption turns out to be 4 times stronger than the El Niño impact in 2015–2016. Consequently, the reanalysis meteorology (here ERA–Interim) in combination with a sophisticated chemistry transport model (here CLaMS) realistically represents the effects of the interplay of QBO and ENSO on lower stratospheric $H_2O$.

Current climate models predict a shift of the basic state toward more frequent El Niño conditions as well as a weakening QBO amplitude in the lower stratosphere for the future climate due to anthropogenic climate change (van Oldenborgh et al., 2005; Timmermann et al., 1999; Cai et al., 2014; Kawatani and Hamilton, 2013). Hence, the interplay of ENSO events and QBO phases affecting the lower stratospheric water vapor and ozone is likely to change, causing changes in radiative forcing of surface climate. An improved understanding of the interplay between ENSO events and QBO phases will help to reduce related uncertainties in climate projections as well as in past and future lower stratospheric $H_2O$ trends (Kunz et al., 2013; Hegglin et al., 2014). In addition, subtle differences in the alignment of ENSO and QBO could contribute to the large spread in basic state cold point tropopause temperature between different climate models and induced ozone radiative feedback (Birner and Charlesworth, 2017; Ming et al., 2017).

## 7 Summary and Conclusions

Based on an established multiple regression method applied to Aura–MLS observations and CLaMS model simulations, we found that both the most recent El Niño and QBO disruption in 2015–2016 induced substantial changes in the lower stratospheric $O_3$ and $H_2O$. The El Niño induced substantial positive anomalies of up to 10% in $H_2O$ and negative anomalies of about 15% in $O_3$. Our results also demonstrated that if there had been no QBO disruption, the lower stratosphere would likely have been substantially moistened by the alignment of the El Niño with the westerly QBO, with deseasonalised anomalies exceeding 25%.

In boreal winter of 2015–2016 (September 2015–March 2016), the alignment of the strong El Niño with the westerly QBO strongly moistened the lower stratosphere (positive anomalies of more than 20%). However, the sudden shift in the QBO from westerly to easterly wind shear reversed the moistening of the lower stratosphere between 380 and 450 K, leading to large negative $H_2O$ anomalies of as much as 20% by autumn 2016 (4 times bigger than the El Niño influence in early 2016). The QBO also led to positive $H_2O$ anomalies over 460–600 K from April to December 2016. The El Niño-induced $H_2O$ anomalies are opposite to the easterly QBO-induced $H_2O$ changes. This opposite response arises because the QBO affects the atmosphere in a zonally symmetric manner, whereas ENSO predominantly creates zonally asymmetric signatures (source region of the

dehydration (Konopka et al., 2016; Avery et al., 2017)), and therefore the two mechanisms give rise to different patterns of variability in the tropical cold point tropopause temperatures. Interestingly, although this QBO shift reversed the moistening of the lower stratosphere, the $O_3$ mixing ratios continued to decrease in the tropics, indicating an additional acceleration of the BD-circulation.

The control of stratospheric $H_2O$ anomalies strongly depends on the interaction of ENSO events and QBO phases. The alignment of the westerly QBO phase with El Niño and easterly QBO phase with La Niña are the key factors regulating the stratospheric water budget. The interaction of El Niño–westerly QBO phase leads to large positive lower stratospheric $H_2O$ anomalies, while the interplay between La Niña and easterly QBO phase leads to negative anomalies. During weak and moderate ENSO events, the $H_2O$ anomalies are controlled by the QBO phase. The effects of QBO and ENSO on lower

stratospheric $H_2O$ in the MLS observations are consistent with CLaMS model results.

       Our results suggest that the interplay of ENSO events and QBO phases will be crucial for the control of the lower strato-spheric water vapor and ozone budget under changing future climate, when increasing El Niño-like conditions (Timmermann et al., 1999; Cai et al., 2014) and a decreasing lower stratospheric QBO amplitude (Kawatani and Hamilton, 2013) are expected. The interplay will change, with ENSO likely controlling the lower stratospheric trace gas variability more strongly in

the future. It is clear that ENSO impacts both tropopause height and tropopause temperature. Future analysis is needed using sensitivity runs from global circulation models and coupled chemistry-climate models to diagnose and separate the impact of future changes in tropopause height and tropopause temperature on stratospheric water.

*Data availability.*  Aura Microwave Limb Sounder product (http://disc.sci.gsfc.nasa.gov/Aura/data-holdings/MLS/index.shtml) and ERA-Interim reanalysis data (https://www.ecmwf.int/en/forecasts/datasets/reanalysis-datasets/era-interim). CLaMS $H_2O$ data set can be requested

from the corresponding author M. Diallo (m.diallo@fz-juelich.de).

*Competing interests.*  The authors declare that they have no conflict of interest.

*Acknowledgements.*  We particularly thank the NASA Jet Propulsion Laboratory and the European Centre for Medium-Range Weather Fore-casts for providing Aura Microwave Limb Sounder product (https://mls.jpl.nasa.gov/) and the ERA-Interim reanalysis data. Work at the Jet Propulsion Laboratory, California Institute of Technology, was done under contract with the National Aeronautics and Space Administration.

This work was funded by the Helmholtz Association under grant number VH-NG-1128 (Helmholtz-Hochschul-Nachwuchsforschergruppe). The authors thank the organising commitee of SPARC Training School on stratosphere-troposphere interactions during which this work was initiated. We sincerely thank Chen Schwartz at the Hebrew University of Jerusalem, Israel and Abebe Kebede Bahir Dar University, Ethiopia for helpful discussions.

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

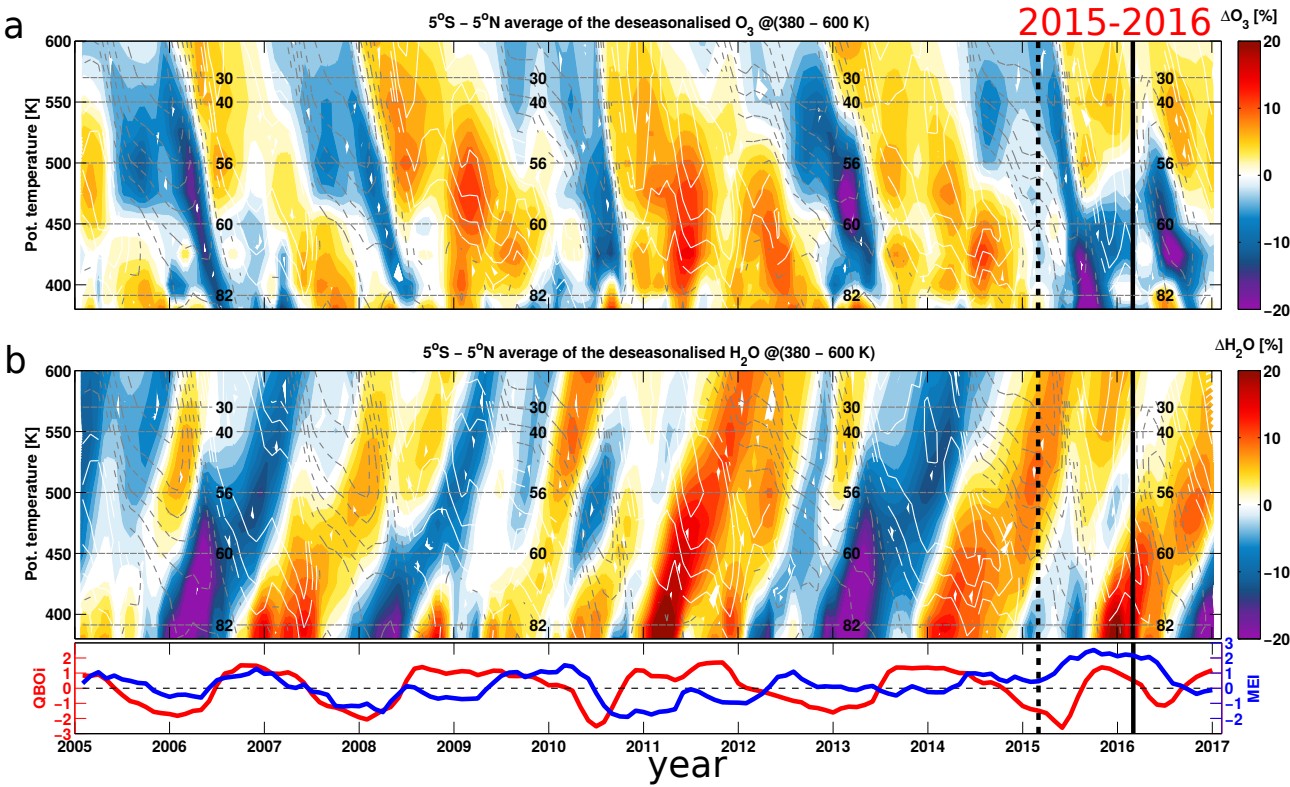

**Figure 1.** Deseasonalised tropical stratospheric $O_3$ and $H_2O$ timeseries from MLS satellite observations for the 2005–2016 period in percent change from long-term monthly means as a function of time and potential temperature. a) Deseasonalised monthly mean $O_3$. b) Deseasonalised monthly mean $H_2O$. Vertical black dashed line indicates February 2015 for the warm ENSO onset. Vertical black line indicates February 2016 for the QBO shift onset. Horizontal gray dashed lines indicate the pressure levels. The lowermost panel shows the QBO index at 50 hPa in red and the MEI index in blue. Monthly averaged zonal mean zonal wind component, u (m·s$^{-1}$), from ERA-Interim is overplotted as solid white (westerly) and dashed grey (easterly) lines.

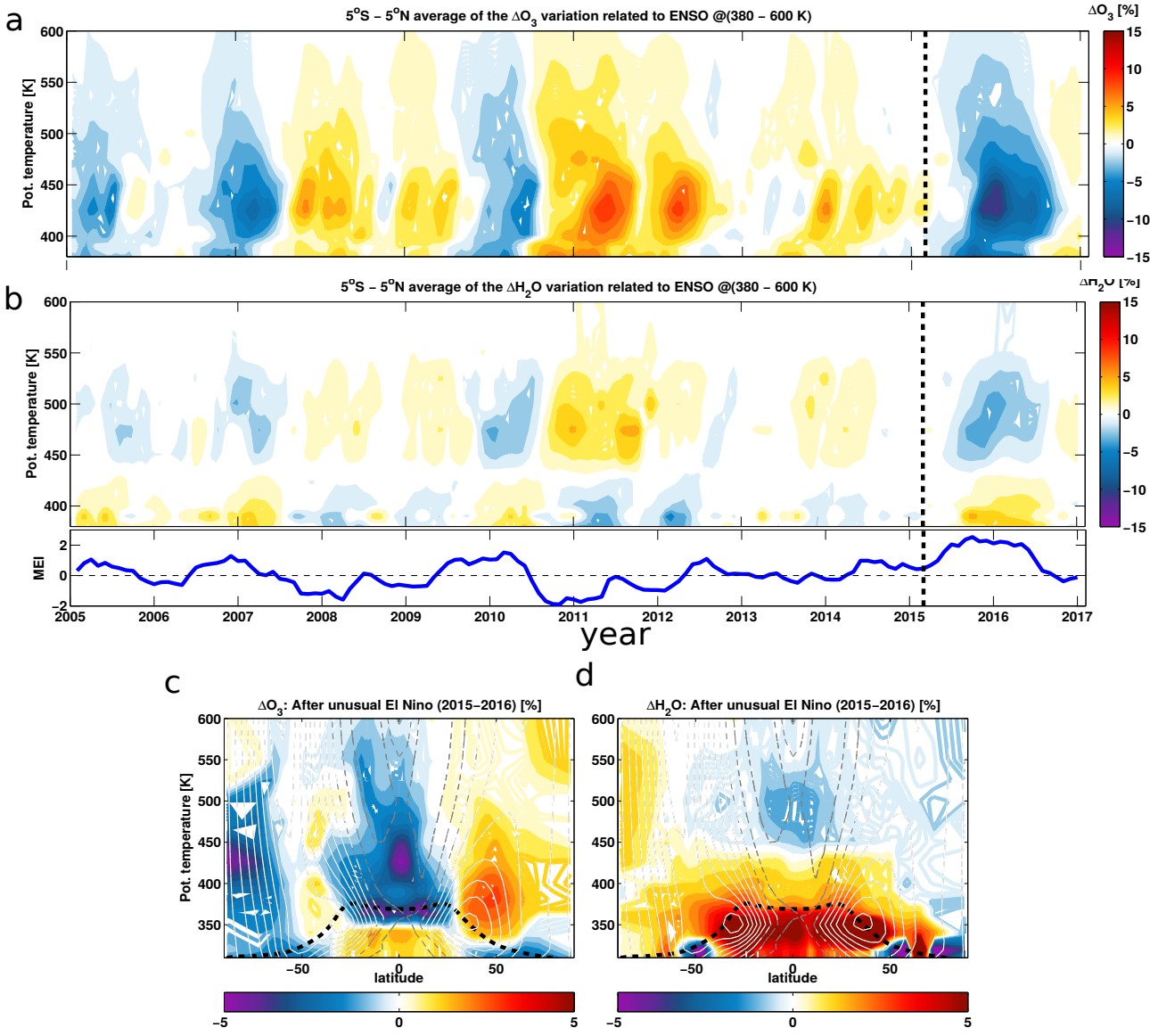

**Figure 2.** ENSO impact on the stratospheric $O_3$ (a) and $H_2O$ (b) from MLS satellite observations for the 2005–2016 period in percent change relative to monthly mean mixing ratio as a function of time and potential temperature. Shown ENSO impact on the stratospheric trace gases is derived from the multiple regression fit as the difference between the residual ($\varepsilon$ in (1)) without and with explicit inclusion of the ENSO signal. Vertical black dashed line indicates the warm ENSO onset (February 2015). The small panel below indicates the MEI index in blue. Panels (c-d) show the zonal distribution of the ENSO impact on stratospheric $O_3$ (c) and $H_2O$ (d) averaged from January 2015–December 2016 in percent change relative to monthly mean mixing ratios. Black dashed horizontal line indicates the tropopause from ERA-Interim. Zonal mean zonal wind component, u (m·s$^{-1}$), averaged over the 2015-2016 period, from ERA-Interim is overplotted as solid white (westerly) and dashed grey (easterly) lines.

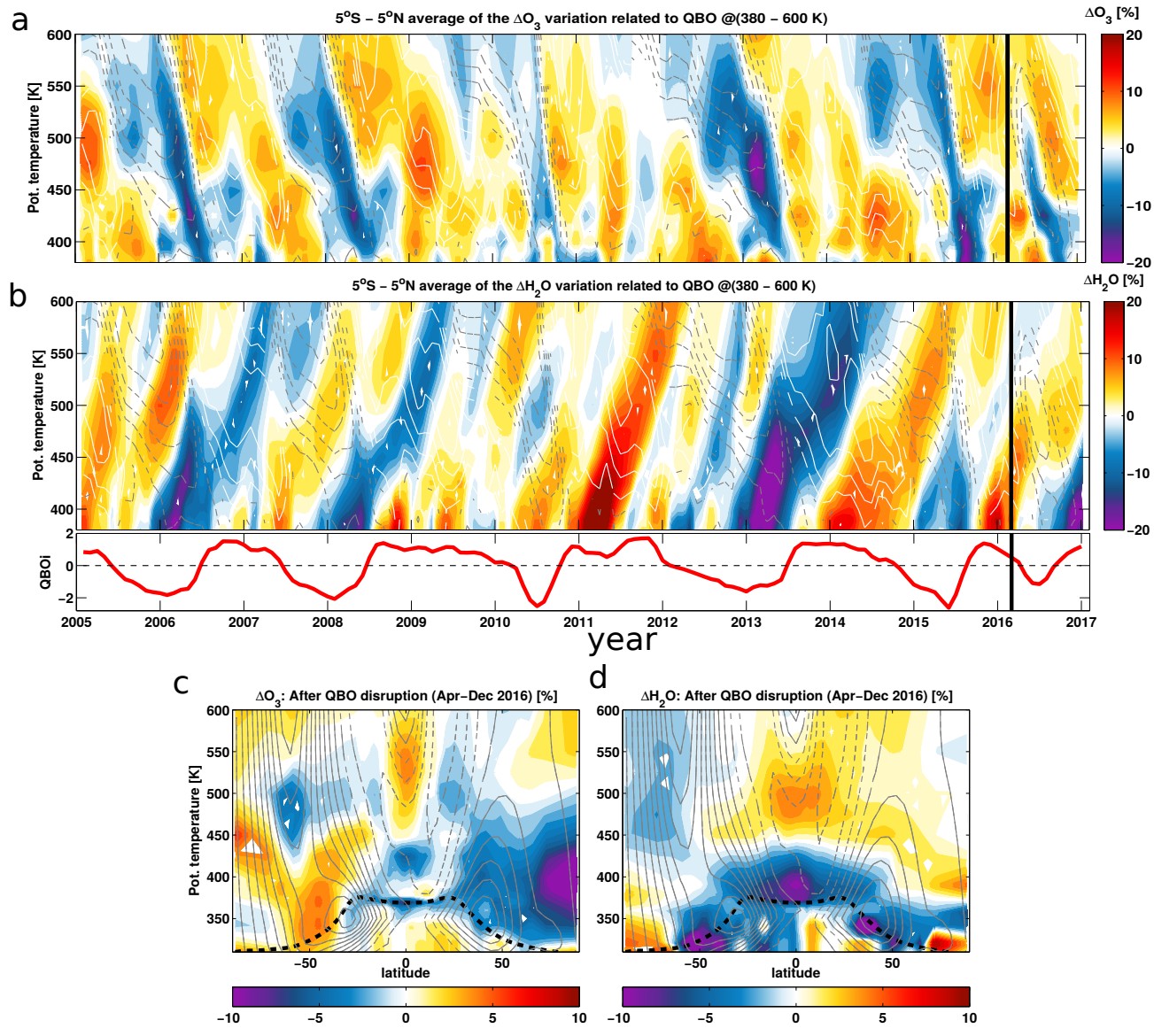

**Figure 3.** QBO impact on the stratospheric $O_3$ (a) and $H_2O$ (b) from MLS satellite observations for the 2005–2016 period in percent change relative to monthly mean mixing as a function of time and potential temperature. Shown QBO impact on the stratospheric trace gases is derived from the multiple regression fit as the difference between the residual ($\varepsilon$ in (1)) without and with explicit inclusion of the QBO signal. Vertical black line indicates the QBO shift onset (February 2016). The small panel below indicates the QBO index at 50 hPa in red. Panels (c-d) show the zonal mean QBO disruption impact on stratospheric $O_3$ (c) and $H_2O$ (d) averaged from April–December 2016 in percent change relative to monthly mean mixing ratios. Black dashed horizontal line indicates the tropopause from ERA-Interim. Monthly mean, zonal mean wind component, u (m·s$^{-1}$), from ERA-Interim is overplotted as solid white (westerly) and dashed grey (easterly) lines.

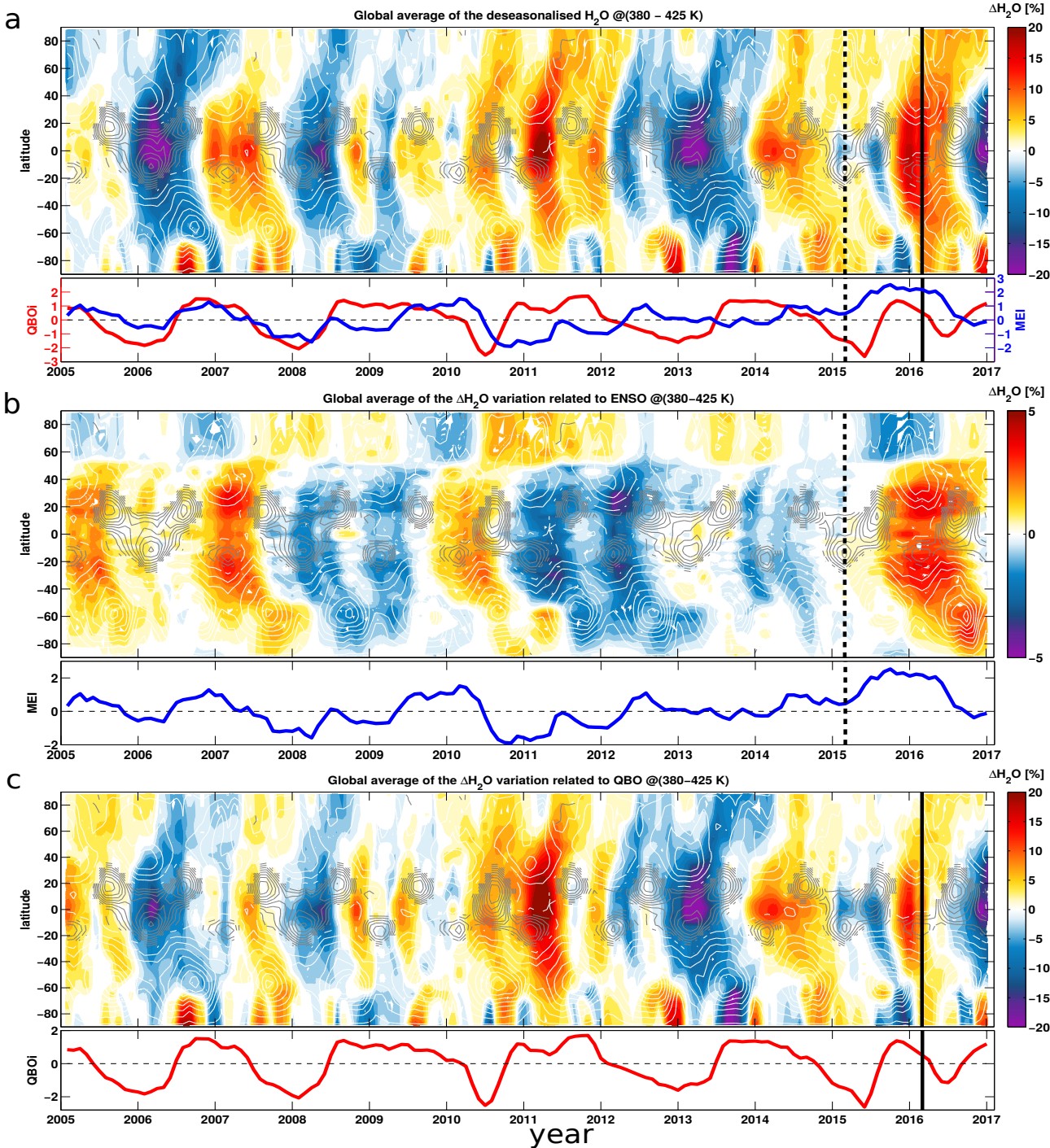

**Figure 4.** Latitude-time evolution of the global deseasonalised MLS $H_2O$ (a) together with the ENSO (b) and QBO (c) impact on lower stratospheric $H_2O$ in percent change from long-term zonal monthly means derived from the multiple regression fit and averaged between 380–425 K for the 2005–2016 period. Note that there is a factor of 4 difference in the color scales in Figs. 4b and 4c reflecting the difference in the magnitude of the $H_2O$ changes related to ENSO compared to those related to the QBO. Vertical black dashed line indicates February 2015 for the warm ENSO onset. Vertical black line indicates February 2016 for the QBO shift onset. Monthly averaged zonal mean zonal wind component, u (m·s$^{-1}$), from ERA-Interim between 380–500 K is overplotted as solid white (westerly) and solid grey (easterly) lines.

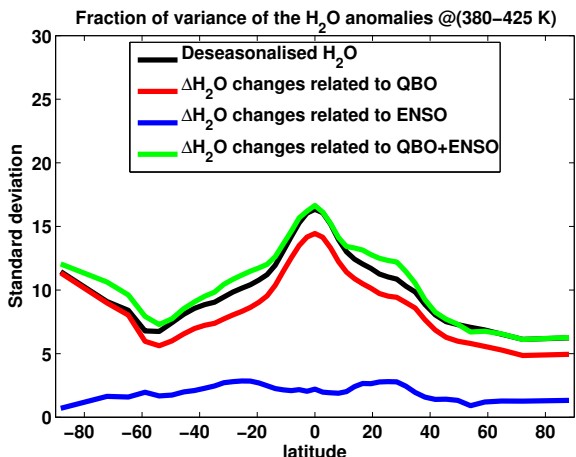

**Figure 5.** Standard deviation (STD) of the global deseasonalised MLS $H_2O$ (black) together with the STD of the ENSO (blue), QBO (red) and ENSO plus QBO (green) impact on lower stratospheric $H_2O$ derived from the multiple regression fit results shown in Fig. 4.

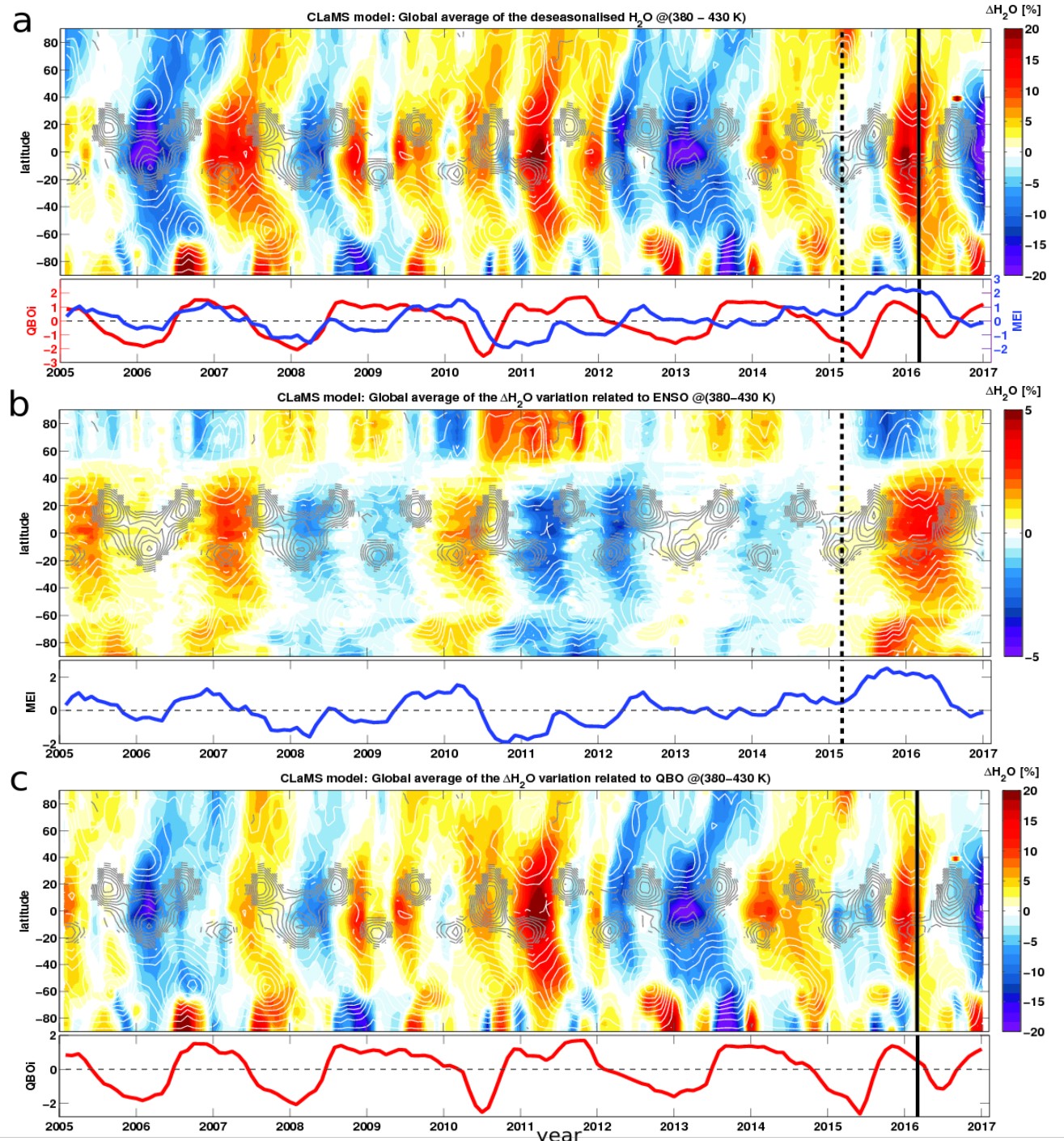

**Figure 6.** Latitude-time evolution of the global deseasonalised CLaMS $H_2O$ (a) together with the ENSO (b) and QBO (c) impact on lower stratospheric $H_2O$ in percent change from long-term zonal monthly means derived from the multiple regression fit and averaged between 380–430 K for the 2005–2016 period. Note that there is a factor of 4 difference in the color scales in Figs. 4b and 4c reflecting the difference in the magnitude of the $H_2O$ changes related to ENSO compared to those related to the QBO. Vertical black dashed line indicates February 2015 for the warm ENSO onset. Vertical black line indicates February 2016 for the QBO shift onset. Monthly averaged zonal mean zonal wind component, $u\,(m\cdot s^{-1})$, from ERA-Interim between 380–500 K is overplotted as solid white (westerly) and solid grey (easterly) lines.