# Peer review of "Response of stratospheric water vapor and ozone to the unusual timing of El Niño and QBO disruption in 2015–2016"

_Atmospheric Chemistry and Physics, 2018_

## Referee Comment (RC1) · Anonymous Referee #1 · 10 May 2018

General Comments: This is an excellent manuscript that addresses the important topic of stratospheric water in present processes and in a changing future climate.

There is a debate in the literature whether the 2015-2016 El Niño event dominated changes to lower stratospheric water [Avery et al., 2017] or whether the QBO disruption played a larger role [Tweedy et al., 2017]. Here, the Diallo et al. manuscript aims to resolve this debate by quantifying the impact of the interaction between the most recent El Niño event and the QBO disruption on stratospheric H2O and O3 from spaceborne measurements during 2015–2016. The paper admits that disentangling the effects of these two processes is challenging, but achieves this quite successfully with a multiple

regression model.

The anomalous stratospheric circulation in boreal winter of 2015–2016 is described. The authors show evidence for the impact of the El Niño event and QBO disruption on stratospheric H2O and O3, and discuss the results in the context of anomalous stratospheric water vapor. In particular, their results suggest that the interplay of ENSO events and QBO phases will be crucial for the control of the lower stratospheric H2O and O3 budgets under changing future climate.

Overall, this is an excellent paper that I recommend for publication after minor revisions (see comments below). The paper is a well-written substantial contribution to scientific progress, with relevant science questions, novel multiple regression analysis tools, and substantial conclusions quantifying the interplay of ENSO and QBO. The results are sufficient to support the interpretations, which have widespread impact. The authors do an excellent job of giving credit, citing the literature, and indicating their own original contribution. I have very few comments because I generally like this manuscript.

Specific comments:

1.1) Page 4, lines 14-15: How accurate are ERA-Interim temperatures in the tropical tropopause layer? Could you please address this?

1.2) Page 4, line 16: the simplified dehydration scheme in CLaMS may be an oversimplification of the microphysical processes that occur in the tropical tropopause layer, controlling water entering the stratosphere. For instance, supersaturation is common in the upper troposphere. Also, stratospheric entry-level value of water vapor is strongly dependent on temperature (see my point 1.1). How do we know that the processes that control stratospheric water are properly represented by the reanalysis meteorology (ERA-Interim) in combination with the CLaMS transport model?

Admittedly, the remarkable similarity of Figures 4 and 5 lends confidence to the CLaMS model, but one should be careful about interpretation because of the MLS vertical

resolution of ∼2.5 to 3 km for H2O. Is this vertical resolution similar to the model, or mismatched with the model?

1.3) Page 6, lines 26-28: given the MLS vertical resolution, how do you separate tropospheric water vapor anomalies from stratospheric anomalies?

1.4) In your conclusion, page 10, lines 30-32, you conclude that your "results suggest that the interplay of ENSO events and QBO phases will be crucial for the control of the lower stratospheric water vapor and ozone budget under changing future climate..." Do you address the separate impact of future changing tropopause height/tropopause temperature on stratospheric water?

Technical corrections (minor): 2.1) Page 3, line 11: I do not like the word "unprecedented" because you actually mean to say "previously unobserved". ENSO, QBO and stratospheric water vapor have only been monitored during the era of satellite observations. A similar interplay between ENSO and QBO could have occurred in the historic or even geologic record.

2.2) Page 8, line 27: I recommend that you replace "unprecedented changes" with "changes larger than previously observed".

2.3) Figure 1: Please define the horizontal black lines in the Figure 1 caption. Are these Pressures?

2.4) Figures 1,2,3,4,5: I recommend that you label the x-axis with "year".

---

## Referee Comment (RC2) · Anonymous Referee #2 · 31 May 2018

Review of "Response of stratospheric water vapor and ozone to the unusual timing of El Nino and QBO disruption in 2015-2016" by Diallo et al.

This study examines the combined impact of the 2015/2016 QBO disruption and El Nino event on lower stratospheric ozone and water vapour concentrations using satellite data. It is clearly written and presented and I believe suitable for publication in ACP pending minor revisions.

Major comments:

1. In Sec. 2, two previous studies examining ENSO and QBO effects on stratospheric water vapour are cited, Avery et al. 2017 and Tweedy et al. 2017, which came to

contrasting conclusions re. the combined roles of ENSO and the QBO. This study has the same goal, and reaches a conclusion that seems closer to Tweedy et al. 2017 (that the QBO had a dominant effect on water vapour following the QBO disruption). But it isn't clearly described how the current study differs in its approach from these previous two. Is it the use of MLS data? The multiple regression approach? Please clarify what is distinct about this study and how it builds on the previous ones. Some more detailed discussion of how the results compare to the previous studies might also be appropriate in the Discussion section.

2. In Sec. 3, random & systematic uncertainties are quoted for the MLS data that seem similar in size to the regression signals reported here (note also the p4, line 5 comment below). It is also noted (p4, line 3) that unrealistic values in the low-latitude UTLS were a problem in previous versions of the MLS data, which sounds worrying since that is the main region of focus in this study. I'm not sure how to compare the reported uncertainties to the regression values. Are these random uncertainties that are applicable to single measurements, such that the regression would effectively beat down the noise? Are there systematic offsets (biases)? More discussion of what these values represent and how they could affect the results would be useful here. It's good that a number of references for data quality are provided (p4, lines 7-9), but a concise explanation of why the regression results in this paper should be believable should also be provided here.

3. In Sec. 4 (p5, line 30) it says that the differencing of residuals gives results similar to direct calculations. In that case, why not just do the direct calculation? Perhaps the lead author's previous work explains this, but a concise explanation should be given here. If there's an advantage in doing it this way, what is it?

Comments by page & line number:

p2, 1: "This moistening" - are you referring to methane oxidation?

p2, 22: On p9, line ~15, you say that easterly shear in the tropical lower stratosphere

speeds up the shallow branch of the Brewer-Dobson circulation. But here you say that westerly shear is associated with enhanced poleward transport. These seem to contradict each other, please clarify.

p2, 26: "A major" –> "Another major"

p2, 30: Here it says that El Nino cools the lower stratosphere, but you go on to say (p3, line 16) that El Nino warms the tropopause. Please add some additional comments here to explain the distinction between the tropopause response to El Nino and lower stratospheric response. As shown in Mitchell et al 2014 (Signatures of naturally induced variability in the atmosphere using multiple reanalysis datasets), Fig 15b shows tropospheric warming and stratospheric cooling with a node near the tropopause. How robust is the tropopause response to ENSO? If tropopause warming is a distinct regional feature of the ENSO response (you go one to discuss the distinction between zonal-mean and regional responses to ENSO), this would be a good place to introduce and describe those differences.

p3, 25: "contains" –> "describes"

p4, 5: Why are O3 uncertainties give as percentages but H2O uncertainties are given in ppmv? Since the figures show O3 and H2O changes in percent, percentages for all these uncertainties would be useful.

p4, 21: Unclear what "properly" means here, suggest delete it.

p5, 5: What does "sorted out" refer to? Please be more specific.

p5, 11: Not sure what "breaking the easterly-westerly phase asymmetry" refers to in this context.

p5, 21: Insert "as expected" before "due to", since the upwelling is not actually observed.

p5, 31: "basis functions" - do you mean the predictor time series (indices)? Please

clarify what is meant here.

p6, 8: "controlling" –> "warming"

p6, 14: Insert "as expected to be" before "due to" (for same reason as the p5, line 21 comment).

p7, 9-10: If the H2O anomalies are delayed, how can they be in phase with the O3 anomalies? Perhaps say "roughly in phase", if this is what you mean.

p7, 13: "by enhancing" –> "consistent with"

p9, p4-5: Based on eyeballing Fig 4, I think I agree. But could this conclusion be made more quantitative, e.g. by saying what is the fraction of variance of the deseasonalized time series that's captured by QBO and ENSO? Or plotting the residual of the full regression in the same style as Fig 4? (Perhaps to include as supplemental so as not to clutter Fig 4.)

p9, 7: "controlled by" –> "dominated by" seems more appropriate to me since the responses here are linear by definition (because multiple linear regression has been used to diagnose them).

p9, 9: "predominated" –> "dominated"

p10, 8: Not clear what "robust" means in this context; suggest delete it.

p10, 18: "led positive" –> "led to positive"

p10, 29: "turn out to be" –> "are"
* * *

---

## Author Comment (AC1) · 20 Jul 2018

**Answer to Reviewers Comments for "Response of stratospheric water vapor and ozone to the unusual timing of El Niño and QBO disruption in 2015–2016" by Mohamadou Diallo et al.**

Dear Editor-in-Chief, Karen Rosenlof,

We are submitting our revised article titled "Response of stratospheric water vapor and ozone to the unusual timing of El Niño and QBO disruption in 2015–2016". We thank the two Reviewers for their detailed and well thought-out comments, which helped to significantly improve the paper. We have made substantial changes to the manuscript in order to thoroughly address the Reviewers' suggestions and comments. Main changes concern:

- an additional new figure 5 describing the fraction of variance of the deseasonalized time series that's captured by QBO and ENSO in the manuscript as suggested by Reviewer #2

- adding a new paragraph in the discussion as suggested by Reviewer #2

- rephrasing of certain paragraphs in order to clarify the manuscript.

With these changes, we are convinced that the paper is highly relevant for a wide-ranging journal like *Atmospheric Chemistry and Physics*. Please see below our answers point by point to all reviewers comments and suggestions.

Reviewers comments are in bold, followed by our respective replies. Changes in the manuscript are in blue, allowing them to be tracked easily.
Kind regards,
Mohamadou Diallo (on behalf of the co-authors)

**Reviewer #1 (Comments to Author):**

**Specific comments:**

1. ***Page 4, lines 14-15: How accurate are ERA-Interim temperatures in the tropical tropopause layer? Could you please address this?***

   We thank Reviewer 1 for her/his instructive suggestion for discussing the accuracy of ERA-I temperatures in the tropical tropopause layer (TTL). ERA-Interim, especially since the 2000s, lies in the middle of all reanalyses in a multi-reanalysis comparison (Long et al. 2017) and is one of the best-performing reanalyses currently available. The assimilation of Global Positioning System radio occultation data since December 2006 have reduced the ERA-Interim cold temperature bias compared with radiosondes in the tropopause layer and the lower stratosphere (Poli et al. 2010). ERA-Interim tropical tropopause temperatures have been shown to compare very well against aircraft measurements in the TTL region over the eastern tropical Pacific (Ueyama et al. 2014). By comparing the ERA-Interim reanalysis with PreConcordiasi balloon observations, Podglajen et al. (2014) find fairly good agreement, with a weak positive bias of 0.6 K and a standard deviation of 1.8 K. These previous studies provide confidence in the use of ERA-Interim temperatures. We have added a discussion about this aspect at Page 5, Lines 7-14.

2. ***Page 4, line 16: the simplified dehydration scheme in CLaMS may be an oversimplification of the microphysical processes that occur in the tropical tropopause layer, controlling water entering the stratosphere. For instance, supersaturation is common in the upper troposphere. Also, stratospheric entry-level value of water vapor is strongly dependent on temperature (see my point 1.1). How do we know that the processes that control stratospheric water are properly represented by the reanalysis meteorology (ERA-Interim) in combination with the CLaMS transport model? Admittedly, the remarkable similarity of Figures 4 and 5 lends confidence to the CLaMS model, but one should be careful about interpretation because of the MLS vertical resolution of 2.5 to 3 km for H2O. Is this vertical resolution similar to the model, or mismatched with the model?***

   We agree with the reviewer that the simplified dehydration scheme in CLaMS may not capture all details about the microphysical processes that occur in the tropical tropopause layer, controlling water entering the stratosphere. Using a trajectory model driven by ERA-Interim and in-situ observations for temperature intercomparison, Schoeberl et al. (2012) show that even small temperature differences between

reanalyses and observations can still induce significant differences in the associated water vapor saturation mixing ratio. However, the CLaMS water vapor anomaly agrees extremely well with time series of MLS-HALOE and Boulder water vapor anomalies as shown in Tao et al. (2015). The climatological tape recorder from CLaMS water vapor has also been compared with MLS (Tao et al. 2015). They find remarkably good agreement between CLaMS and MLS in terms of mean vertical structure and the slow upward phase propagation. This good agreement between CLaMS and observations (Ploeger et al. 2013, Tao et al. 2015, Lossow et al. 2018) gives good confidence in the CLaMS water vapor reconstruction from the large-scale perspective. As pointed out by the reviewer, the remarkable similarity of Figures 4 and 5 is a good example. For the wind and temperature fields, CLaMS uses the native ERA-I vertical resolution, therefore, has higher vertical resolution than MLS. The mean vertical resolution of air parcels in CLaMS Lagrangian model is about 400m near the tropopause. We have added a discussion about this aspect at page 4-5, Lines 34,1-4 and page 5, Lines 14-19. We have also acknowledged the possible effect of the MLS vertical resolution of 2.5 to 3 km for H2O at page 7, lines 27-30 and in next question.

3. ***Page 6, lines 26-28: given the MLS vertical resolution, how do you separate tropospheric water vapor anomalies from stratospheric anomalies?***

    Concerning the MLS vertical resolution, there is not much we can do about it as we use the latest version 4.2, which is provided to users by the MLS team and used in the previous studies (Tweedy et al. 2017; Avery et al. 2017). However, we discussed in the early manuscript (page 7, line 27-30) the possible effect of the smearing arising from the 2.5-3 km vertical resolution of the MLS water vapor measurements on the positive anomalies near the tropopause region.

4. ***In your conclusion, page 10, lines 30-32, you conclude that your "results suggest that the interplay of ENSO events and QBO phases will be crucial for the control of the lower stratospheric water vapor and ozone budget under changing future climate . . . " Do you address the separate impact of future changing tropopause height/tropopause temperature on stratospheric water?***

    In this last sentence of the conclusion, we speculate based on current knowledge in the literature and findings that the interplay will be crucial for the control of the lower stratospheric water vapor and ozone budget under changing future climate. In a changing future climate with a projection of a decreasing lower stratospheric QBO amplitude (Kawatani et al. 2013) combined with increasing frequency of El Niño-like conditions (Cai et al. 2014), the interplay will change with ENSO, hence likely controlling the lower stratospheric trace gas variability more strongly in the future. It is clear that ENSO impacts both tropopause height/tropopause temperature. Future analysis is needed using CCMs/GCM sensitivity runs to diagnose and separate the impact of future changing tropopause height/tropopause temperature on stratospheric water. We have added more details at Page 11-12, last lines in sect. 7.

**Technical corrections (minor):**

1. ***2.1) Page 3, line 11: I do not like the word "unprecedented" because you actually mean to say "previously unobserved". ENSO, QBO and stratospheric water vapor have only been monitored during the era of satellite observations. A similar interplay between ENSO and QBO could have occurred in the historic or even geologic record.***

    We have rephrased it.

2. ***Page 8, line 27: I recommend that you replace "unprecedented changes" with "changes larger than previously observed".***

    We have rephrased it.

3. ***Figure 1: Please define the horizontal black lines in the Figure 1 caption. Are these Pressures?***

    Thanks for the remark. Yes these black lines indicate isobars lines. We have defined the horizontal black lines now in the captions.

4. ***Figures 1,2,3,4,5: I recommend that you label the x-axis with "year".***

    We have now labelled the x-axis with "year" in these figures.

---

## Author Comment (AC2) · 20 Jul 2018

**Answer to Reviewers Comments for "Response of stratospheric water vapor and ozone to the unusual timing of El Niño and QBO disruption in 2015–2016" by Mohamadou Diallo et al.**

Dear Editor-in-Chief, Karen Rosenlof,

We are submitting our revised article titled "Response of stratospheric water vapor and ozone to the unusual timing of El Niño and QBO disruption in 2015–2016". We thank the two Reviewers for their detailed and well thought-out comments, which helped to significantly improve the paper. We have made substantial changes to the manuscript in order to thoroughly address the Reviewers' suggestions and comments. Main changes concern:

- an additional new figure 5 describing the fraction of variance of the deseasonalized time series that's captured by QBO and ENSO in the manuscript as suggested by Reviewer #2

- adding a new paragraph in the discussion as suggested by Reviewer #2

- rephrasing of certain paragraphs in order to clarify the manuscript.

With these changes, we are convinced that the paper is highly relevant for a wide-ranging journal like *Atmospheric Chemistry and Physics*. Please see below our answers point by point to all reviewers comments and suggestions.

Reviewers comments are in bold, followed by our respective replies. Changes in the manuscript are in blue, allowing them to be tracked easily.
Kind regards,
Mohamadou Diallo (on behalf of the co-authors)

**Anonymous Referee #2:**

**Major comments:**

1. ***In Sec. 2, two previous studies examining ENSO and QBO effects on stratospheric water vapour are cited, Avery et al. 2017 and Tweedy et al. 2017, which came to contrasting conclusions i.e. the combined roles of ENSO and the QBO. This study has the same goal, and reaches a conclusion that seems closer to Tweedy et al. 2017 (that the QBO had a dominant effect on water vapour following the QBO disruption). But it isn't clearly described how the current study differs in its approach from these previous two. Is it the use of MLS data? The multiple regression approach? Please clarify what is distinct about this study and how it builds on the previous ones. Some more detailed discussion of how the results compare to the previous studies might also be appropriate in the Discussion section.***

   The two previous studies (i.e. Avery et al. (2017) and Tweedy et al. (2017)) focus on ENSO and QBO, respectively. Combining observations and trajectory calculations with a simple linear regression analysis, Avery et al, (2017) focus on mainly the impact of ENSO on lower stratospheric water vapor and on convective cloud ice. They did mention that a "simple regression of a 70 hPa shear-based QBO index with MLS water vapour data at 82 hPa shows that the QBO can account for only 0.1 ppmv of the observed zonal mean water vapour anomaly", but the QBO impact on lower stratospheric water vapor is largely overlooked in this study. Regarding Tweedy et al. (2017), they focus on the impact of QBO disruption on lower stratospheric water vapor using MLS and MERRA data combined with a composite (correlation) analysis. Even though, we use similar observations of water vapor and ozone from MLS, our study differs from these two studies in terms of the method. We use a multiple regression with a lag that takes into account the history of the predictors. Therefore, our regression approach carefully disentangles the impact of ENSO and QBO signals on lower stratospheric trace gases, including water vapor and ozone. In addition, our analysis focus on the global lower stratospheric water vapor response to ENSO and QBO (310-600K), while the two previous studies mainly focus on tropical water vapor time series response at 82 hPa. We have added more specific comments about the method used in the two previous studies in section 2 and a new paragraph recapping the studies of Tweedy et al. (2017) and Tweedy et al (2017) in the discussion section.

2. ***In Sec. 3, random and systematic uncertainties are quoted for the MLS data that seem similar in size to the regression signals reported here (note also the p4, line 5 comment below). It is also noted (p4, line 3) that unrealistic values in the low-latitude UTLS were a problem in previous versions of the MLS data, which sounds worrying since that is the main region of focus in this study. I'm not sure how to compare the reported uncertainties to the regression values. Are these random uncertainties that are applicable to single measurements, such that the regression would effectively beat down the noise? Are there systematic offsets (biases)? More discussion of what these values represent and how they could affect the results would be useful here. It's good that a number of references for data quality are provided (p4, lines 7-9), but a concise explanation of why the regression results in this paper should be believable should also be provided here.***

We were not precise about the unrealistic values. According to Livesey et al. (2008), these unrealistic values in the tropics were a problem at 215 hPa in MLS v2.2 for $O_3$ and CO data, which is below the region of interest in this study. We have corrected it in the manuscript. The reported systematic uncertainties are provided by MLS quality control report of the version 4.2 data set (Livesey et al. 2017; Santee et al. 2017) and are applicable to individual profile measurements with a spatial representativeness of 200-300 km along the orbital-track line of sight. The regression results are not affected as these intrinsic uncertainties are applied on the $H_2O$ and $O_3$ mixing ratios and not the anomalies as the random errors are not relevant for the averaged monthly zonal mean $H_2O$ anomalies used in this study. Hegglin et al. 2013 show that Aura-MLS zonal monthly mean $H_2O$ data show very good to excellent agreement with the Multi-Instrument Mean (MIM) in comparison between 13 instruments, throughout most of the atmosphere (and also throughout tropical UTLS) with mean deviations from the MIM between +2.5 and +5% only. In addition, MLS data is one of the best global coverage observations that we currently get and is widely used. We have corrected it in the manuscript (Page 4, Lines 20-26).

3. ***In Sec. 4 (p5, line 30) it says that the differencing of residuals gives results similar to direct calculations. In that case, why not just do the direct calculation? Perhaps the lead author's previous work explains this, but a concise explanation should be given here. If there's an advantage in doing it this way, what is it?***

There's not a specific advantage to doing it in this way. We have previously proven that the two methods are consistent (Diallo et al. 2017), therefore, we use it as we do not need to reconstruct the time series after the regression.

**Comments by page & line number:**

1. ***p2, 1: "This moistening" - are you referring to methane oxidation?***

"This moistening" refers to the moisture transport from the troposphere into the stratosphere. We have rephrased it.

2. ***p2, 22: On p9, line 15, you say that easterly shear in the tropical lower stratosphere speeds up the shallow branch of the Brewer-Dobson circulation. But here you say that westerly shear is associated with enhanced poleward transport. These seem to contradict each other, please clarify.***

The QBO modulation of the BDC includes enhanced tropical upwelling during the easterly phase, while the westerly phase reduces the tropical upward motion, but enhances the horizontal mixing and transport toward the polar regions (e.g. Plumb and Bell, 1982; Trepte and Hitchman, 1992). We have rephrased it to improve the clarity.

3. ***p2, 26: "A major" –> "Another major"***

We have rephrased it.

4. ***p2, 30: Here it says that El Nino cools the lower stratosphere, but you go on to say (p3, line 16) that El Nino warms the tropopause. Please add some additional comments here to explain the distinction between the tropopause response to El Nino and lower stratospheric response. As shown in Mitchell et al 2014 (Signatures of naturally induced variability in the atmosphere using multiple reanalysis datasets), Fig 15b shows tropospheric warming and stratospheric cooling with a node near the tropopause. How robust is the tropopause response to ENSO? If tropopause warming is a distinct regional feature of the ENSO response (you go one to discuss the distinction between zonal-mean and regional responses to ENSO), this would be a good place to introduce and describe those differences.***

We have rephrased these sentences and added a discussion of the distinct ENSO effect on the zonal mean and regional temperature anomalies. Please see page 2, lines 34-35 and page 3, lines 1-4.

5. **p3, 25: "contains" –> "describes"**

We have rephrased the sentence.

6. **p4, 5: Why are O3 uncertainties give as percentages but H2O uncertainties are given in ppmv? Since the figures show O3 and H2O changes in percent, percentages for all these uncertainties would be useful.**

The O3 and H2O uncertainties are just the values provided in the literature by the MLS team as reported in Santee et al., 2017.

7. **p4, 21: Unclear what "properly" means here, suggest delete it.**

We have deleted it.

8. **p5, 5: What does "sorted out" refer to? Please be more specific.**

"sorted out" meant "selected from a range of possible values". We have rephrase it.

9. **p5, 11: Not sure what "breaking the easterly-westerly phase asymmetry" refers to in this context.**

We mean the breaking of the regular cycle of easterly-westerly. We have rephrased it.

10. **p5, 21: Insert "as expected" before "due to", since the upwelling is not actually observed.**

We have rephrased it.

11. **p5, 31: "basis functions" - do you mean the predictor time series (indices)? Please clarify what is meant here.**

Yes, we mean by basis function the ENSO and QBO predictor time series. We have rephrased it.

12. **p6, 8: "controlling" –> "warming"**

We have rephrased it.

13. **p6, 14: Insert "as expected to be" before "due to" (for same reason as the p5, line 21 comment).**

We have inserted it.

14. **p7, 9-10: If the H2O anomalies are delayed, how can they be in phase with the O3 anomalies? Perhaps say "roughly in phase", if this is what you mean.**

We have rephrased it.

15. **p7, 13: "by enhancing" –> "consistent with"**

We have rephrased it.

16. **p9, 4-5: Based on eyeballing Fig 4, I think I agree. But could this conclusion be made more quantitative, e.g. by saying what is the fraction of variance of the deseasonalized time series that's captured by QBO and ENSO? Or plotting the residual of the full regression in the same style as Fig 4? (Perhaps to include as supplemental so as not to clutter Fig 4.)**

Thanks for this good suggestion. We have added a new figure describing the fraction of variance of the deseasonalized time series that's captured by QBO and ENSO in the manuscript. This figure confirms that the dominant part of variability in the water vapor anomalies is induced by QBO and ENSO, with QBO contributing the largest part. Please see Page 9, lines 21-25.

17. **p9, 7: "controlled by" –> "dominated by" seems more appropriate to me since the responses here are linear by definition (because multiple linear regression has been used to diagnose them).**

The analysis is based on multiple regression with a lag, allowing the history of the predictors to be taken into account in a manner not possible with a simple linear regression. The method becomes linear only after the optimal lag is found. We have rephrased it.

18. **p9, 9: "predominated" –> "dominated"**

We have rephrased it.

19. ***p10, 8: Not clear what "robust" means in this context; suggest delete it.***
    We have rephrased it.

20. ***p10, 18: "led positive" –> "led to positive"***
    We have rephrased it.

21. ***p10, 29: "turn out to be" –> "are"***
    We have rephrased it.

---

## Author Response (AR2)

**Answer to Reviewers Comments for "Response of stratospheric water vapor and ozone to the unusual timing of El Niño and QBO disruption in 2015–2016" by Mohamadou Diallo et al.**

Dear Co-editor, Karen Rosenlof,

We are submitting our revised article titled "Response of stratospheric water vapor and ozone to the unusual timing of El Niño and QBO disruption in 2015–2016". We thank the Co-editor for her minor comments. We have made these changes to the manuscript in order to address the Co-editor' suggestions and comments. Please see below our answers point by point to the Co-editor comments and suggestions. Co-editor comments are in bold, followed by our respective replies. Changes in the manuscript are in blue, allowing them to be tracked easily.

Kind regards,
Mohamadou Diallo (on behalf of the co-authors)

**Technical corrections (minor revisions):**

1. ***Overall, the response has been adequate. I have 2 requests before publishing. The first is that the paper be looked at carefully for grammar. The second is that the authors consider the following comments.***

   We appreciate the careful reading of the manuscript and comments. We tried to address it.

2. ***This sentence needs to be rewritten: page 2: line 28-30 "The tropical upwelling is anti-correlated with the tropical temperature above the tropopause and its strength determines stratospheric ozone and water vapor entry values by modulating TTL temperatures (Yulaeva et al., 1994; Randel et al., 2006; Flury et al., 2013)." Note, modulating TTL temperatures doesn't determine stratospheric ozone. Rewrite to ensure that it is clear that advection of tropospheric ozone into the stratosphere is the important process for ozone.***

   Thanks for pointing this unclear formulation out. We agree that the modulation of TTL temperatures does not determine stratospheric ozone. The advection of tropospheric ozone into the stratosphere does not also control the stratospheric ozone. The advection acts locally, so local upwelling strength will be anti-correlated with ozone, whether or not we still have a tropospheric signature. We have rephrased it.

3. ***Page 3: lines 0-6 (and a few lines at the end of page 2) I think the authors should also consider a recent paper by Chaim Garfinkel (that is not currently referenced...see https://www.atmos-chem-phys.net/18/4597/2018/acp-18-4597-2018.pdf). This considers the Nonlinear response of tropical lower-stratospheric temperature and water vapor to ENSO, and given the magnitude of the El Niño discussed, it may play a role in the water vapor response.***

   We have added the citation.

4. ***Also, Page 3, first paragraph of section 2. The authors should also acknowledge that the Avery paper and the Tweedy paper are really looking at different times. The Avery paper is concerned with the ENSO peak, which is before the QBO disruption.***

   We agree that Avery et al. (2017) mainly focus on December 2015 (boreal winter 2015-2016). However, Tweedy et al. (2017) did considered the same MLS period as in our study. According to Newman et al., (2016), the QBO disruption began developing in December 2015 and was fully complete by mid-April 2016 as shown in their Fig. 3. While, the El Niño was peaking in boreal winter 2015-2016, the QBO disruption was already triggered (Osprey et al., 2016; Newman et al., 2016; Dunkerton, 2016; Coy et al., 2017). Clearly, the impact of the QBO disruption and the El Niño was most effectively when they reach their maximum in mid-April 2016 and boreal winter 2015-2016, respectively, but that does not exclude the fact that the impact of both phenomena was overlapping. As shown on our Fig. 1, El Niño started decaying at later boreal winter 2015-2016, consistent with observed positive $H_2O$ anomalies up to early autumn 2016 in our Fig. 4b. Avery et al., (2017) mainly focused on ENSO and discussed the extremely high water, when the QBO disruption began developing in December 2015. However, they also pointed out that QBO have only small effect on $H_2O$ anomalies, even though they did not mention the QBO

disruption (please see their section "Trajectory modelling" in Avery et al., (2017) as well as figure in the supplement). We have rephrased the respective text in order to clarify these issues.

5. ***On page 5 it states "In addition, the H2O anomaly time series is less affected by biases, which do not have ENSO or QBO signals." This needs a reference or a demonstration that it is true.***

   The biases, as provided by the MLS team (Livesey et al., 2017) only affect the absolute value as it is just a number that one should add/remove from the absolute value but they do not affect the anomalies. We have rephrased it.

6. ***Page 6 states "Together, the most recent El Niño event and the QBO disruption are expected to impact the tropical upwelling, via wavemean flow interaction (Holton, 1979; Dunkerton, 1980; Grimshaw, 1984) and control of the cold point temperatures (Kim and Son, 2012; Kim and Alexander, 2015), therefore affecting the transport and distribution of stratospheric trace gases (Avery et al., 2017; Tweedy et al., 2017)" You need to acknowledge that the peak of the two events (El Nino and QBO disruption) are at different times, so although they both impact tropical upwelling, they don't do it simultaneously.)***

   We have rephrased the respective text in the manuscript.

7. ***Discussion: first sentence (line 10 page 9) doesn't make sense "Two previous studies (i.e. Avery et al. (2017) and Tweedy et al. (2017)) focussing on ENSO and QBO, respectively, led to puzzling H2O anomalies in 2015-2016." The studies did not lead to anomalies.***

   Thank you for pointing this erroneous formulation out. Avery et al. (2017) and Tweedy et al. (2017) made contradictory statements about the H2O anomalies in the lower stratosphere. We have rephrased it.

8. ***The next sentence, does not correctly represent the Avery study, which did not discuss the qbo disruption and is mainly discussing the Dec 2015 extremely high water and attributing that to ice input. The Tweedy paper is only concerned with the qbo disruption that is in 2016, when the ONI was decreasing.***

   We agree that Avery study did not discuss the disruption. We have rephrased the sentence.

[revised manuscript text omitted]